# Extracellular miRNAs for the Management of Barrett’s Esophagus and Esophageal Adenocarcinoma: A Systematic Review

**DOI:** 10.3390/jcm10010117

**Published:** 2020-12-31

**Authors:** Kazumi Inokuchi, Takahiro Ochiya, Juntaro Matsuzaki

**Affiliations:** 1Division of Gastroenterology and Hepatology, Department of Internal Medicine, Keio University School of Medicine, 35 Shinanomachi, Shinjuku-ku, Tokyo 160-8582, Japan; happy.kazooo@keio.jp; 2Department of Molecular and Cellular Medicine, Tokyo Medical University, 6-7-1 Nishishinjuku, Shinjuku-ku, Tokyo 160-0023, Japan; tochiya@tokyo-med.ac.jp

**Keywords:** esophageal cancer, GERD, miRNA, biomarker

## Abstract

Esophageal adenocarcinoma (EAC), the major histologic type of esophageal cancer (EC) in Western countries, is a disease with a poor prognosis, primarily due to usual diagnosis at an advanced stage. The prevalence of EAC has increased in recent years, both in Western countries and in Asia. Barrett’s esophagus (BE) is a precursor lesion of EAC. Therefore, early detection and proper management of BE and EAC is important to improve prognosis. Here, we systematically summarize current knowledge about the potential utility of extracellular microRNAs (miRNAs), which are thought to be non-invasive biomarkers for many diseases, for these purposes. A search of the PubMed and Embase databases identified 22 papers about extracellular miRNAs that have potential utility for management of EAC. Among them, 19 were EAC-related and ten were BE-related; some of these dealt with both conditions. The articles included studies reporting diagnosis, prognosis, and treatment responses. Multiple papers report dysregulation of miR-194-5p in BE and miR-21-5p, -25-3p, and -93-5p in EAC. Although it will take time to utilize these miRNAs in clinical practice, they are likely to be useful non-invasive markers in the future.

## 1. Introduction

Esophageal adenocarcinoma (EAC) is a major histologic type of esophageal cancer (EC), the incidence of which varies geographically according to race and gender [1]. The rate of EAC in Western Europe, Australia, and North America is increasing rapidly [2,3,4]. By 2030, 1 in 100 men in the Netherlands and United States are predicted to be diagnosed with EAC during their lifetime [5]. In Japan, although EAC accounts for less than 10% of EC cases, the incidence is increasing [6]. The increase can be explained by decreased prevalence of *Helicobacter pylori* (*H. pylori*) infection, although the relationship between *H. pylori* and EAC occurrence remains controversial. [7,8,9].

The prognosis of EAC is poor if diagnosed at the advanced stage, by which time metastasis or invasion has occurred. The 5-year survival rate for EAC patients is 80.5% at stage I, and 45.1%, 17.6%, and 2.1% at stages II, III, and IV, respectively [3]. However, increased rates of early detection in European countries, the United States, and China have improved the 5-year survival rate from less than 5% in the 1960s to about 20% in 2000 [10,11,12]. Thus, early identification of EAC and identification of high-risk groups are important.

Barrett’s esophagus (BE) denotes metaplasia from squamous epithelium into columnar epithelium; BE is classified into short-segment BE (SSBE) and long-segment BE (LSBE). BE is considered to be a precancerous lesion of EAC. Both SSBE and LSBE carry a risk of carcinogenesis [13], although the risk is lower for SSBE [14]. BE and EAC are induced by reflux of both gastric and bile acids [15], irrespective of the severity of reflux symptoms and esophagitis [4,16]. Similar to the adenoma-carcinoma sequence of colorectal cancer, EAC develops through metaplasia, low-grade dysplasia (LGD), high-grade dysplasia (HGD), intraepithelial adenocarcinoma, and finally invasive adenocarcinoma. Jason et al. reported that BE develops in 6–14% of patients with GERD, and that 0.5–1% of these will develop adenocarcinoma [4,17]. Thus, to facilitate early detection of EAC, it is important to identify patients with BE who are at high risk of EAC.

Esophagogastroduodenoscopy is the standard method for early detection of BE and EAC; no non-invasive and reliable screening method has been established to date. Although various efforts and trials, such as Seattle protocol, web-based teaching tools [18], and computer-aided detection [19], have been made to improve detection of BE and EAC, the examination itself is invasive and relies heavily on the skill of the operator. The Cytosponge, a gelatin-coated sponge attached to a string that collects cytologic specimens from the esophageal mucosa when withdrawn, may have the potential to replace traditional endoscopic screening and be more cost-effective [20]. However, it is not used widely.

Recently, circulating cell-free DNA and RNA have attracted attention as new biomarkers. MiRNA is a single-stranded non-coding RNA comprising 17–25 bases, which regulates protein translation. It is suggested that tumor tissues may secrete aberrant miRNAs to create a microenvironment conducive to growth. Furthermore, miRNAs are actively secreted into blood, urine, and saliva by extracellular vesicles. Thus, miRNAs may be novel biomarkers for various pathological conditions, including not only cancer but also inflammatory and infectious diseases [21]. Since the discovery of extracellular miRNAs, many studies have actively investigated BE and EAC-specific miRNAs for diagnostic purposes. Here, we systematically reviewed the literature regarding extracellular miRNAs related to BE and EAC.

## 2. Experimental Section

### Methods

The PubMed and Embase databases were searched using the terms “(esophagus OR esophageal) AND (Barrett’s OR adenocarcinoma) AND (microRNA OR miRNA) AND (circulating OR serum OR plasma OR urine)”. Of these, duplications, review articles, studies of intracellular miRNAs only, and studies not investigating BE or EAC (i.e., investigations of esophageal squamous cell carcinoma only) were excluded (Figure 1). The contents of these papers were organized and are described below.

## 3. Results

We identified 35 reports in PubMed and 78 reports in Embase. After applying the exclusion criteria, we identified 22 papers (including abstracts for conference presentations). We classified these into those related to diagnosis of BE, those related to diagnosis of EAC, those related to therapeutic effects, and those related to prognosis of EAC (Figure 2).

### 3.1. Detection of BE

Six studies reported diagnosis of GERD and BE; some of these also mentioned EAC. Those comparing EAC and BE are described in the next section. Here, we describe studies that investigated miRNAs that distinguish individuals with GERD or BE from normal healthy subjects (Table 1).

In 2013, Bansal et al. [22] compared serum levels of miRNAs in 20 BE patients and 19 patients with GERD (grade B or higher). Compared with the GERD group, the BE group had significantly higher levels of miR-15a-5p (sensitivity, 75%; specificity, 75%; area under the receiver operating characteristic curve (AUC), 0.68 (0.60–0.86, *p* = 0.01)) and miR-196a-5p (sensitivity, 70%; specificity, 74%; AUC, 0.72 (0.61–0.89, *p* = 0.03)). A combination of miR-15a and miR-196a can be used to distinguish BE patients from GERD patients, with a sensitivity of 75%, a specificity of 85%, and an AUC of 0.79 (0.64–0.95, *p* = 0.003).

In 2016, Bus et al. [23] identified four BE-specific miRNAs by comparing plasma miRNAs from BE and EAC with those from healthy controls. Further validation in 115 patients and controls showed that miRNA-194-5p and miRNA-451a were up-regulated, and miRNA-136-5p was down-regulated, in BE compared with controls. Furthermore, a combination of three or more miRNAs showed good diagnostic performance, with an AUC of 0.832.

In the same year, Cabbi et al. [24] compared miRNA expression in serum and esophageal mucosal biopsy tissues from patients with BE, columnar-lined esophagus (CLE), and esophagitis. Similar to the results from tissues, circulating levels of miR-143-3p, -194-5p, and -215-5p were up-regulated in the serum of patients with BE, and were higher than those in patients with CLE or esophagitis. MiR-194-5p and miR-215-5p were also significantly up-regulated in patients with CLE compared with those with esophagitis, suggesting that CLE is an intermediate condition between BE and esophagitis.

In 2017, Fassan et al. [25] investigated expression of plasma miR-223-3p in BE and EAC, and found that expression of miR-223-3p was significantly associated with the degree of intestinal metaplasia in those with atrophic gastritis and LSBE.

Pavlov et al. [26] compared serum miRNAs from patients with normal squamous epithelium (SE), BE, HGD, and EAC. MiR-320e was significantly down-regulated in the BE group compared with the SE (*p* ≤ 0.001; AUC, 0.790) and HGD (*p* ≤ 0.005; AUC, 0.786) groups, and miR-199a-3p was significantly down-regulated compared with the SE group (*p* ≤ 0.001; AUC, 0.813).

In 2019, Wang et al. [27] found higher levels of serum miR130a-3p in patients with low-grade and high-grade dysplastic BE than in serum from patients with metaplastic epithelium.
jcm-10-00117-t001_Table 1Table 1MiRNAs used to detect BE.Author, Year [Reference]Sample SizeSampleUp-RegulatedDown-RegulatedBansal et al.2013 [22]BE (*n* = 20)GERD (*n* = 19)Serum-15a-5p-195a-5p
Bus et al.2014 [28]BE (*n* = 8)Healthy (*n* = 6)Plasma-194-5p, -451a-136-5pCabbi et al.2016 [24]BE (*n* = 8)CLE (*n* = 12)Esophagitis (*n* = 10)Plasma-143-3p-215-5p-194-5p
Pavlov et al.2018 [26]BE (*n* = 27)SE (*n* = 19)HGD (*n* = 17)Serum
-320e-199a-3pWang et al. 2019 [27]BE (*n* = 6)Healthy (*n* = 6)Serum-130a-3p
BE, Barrett’s esophagus; CLE, columnar-lined esophagus; SE, squamous epithelium; HGD, high-grade dysplasia.

### 3.2. Detection of EAC

There were 14 publications about the utility of miRNAs for diagnosis of EAC. The controls were mostly healthy individuals or BE patients, and one report used saliva samples (Table 2).

In 2011, Song et al. [29] extracted total RNA from serum samples obtained from EAC patients and healthy controls, and from cancer tissues and normal tissues of EAC patients. Some miRNAs showed significant differences in serum levels between the EAC and healthy groups (AUC, 0.885; sensitivity, 92%; specificity, 83%). They reported that measurement of serum miRNAs is likely to lead to early diagnosis of EAC and to an improved prognosis (the specific name of miRNA was not reported).

In 2013, Xie et al. [30] analyzed saliva miRNAs from EC patients and healthy subjects, and verified their presence in both whole saliva and saliva supernatant samples from 39 EC patients and 39 healthy controls. MiR-10b-3p, -144-3p, and -451a in whole saliva, and miR-10b-3p, -144-3p, -21-5p, and -451a in saliva supernatant, were significantly up-regulated in EC patients (sensitivity, 89.7%, 92.3%, 84.6%, 79.5%, 43.6%, 89.7%, and 51.3%, respectively, and specificity, 57.9%, 47.4%, 57.9%, 57.9%, 89.5%, 47.4%, and 84.2%, respectively). Although this report also included SCC, the authors reported no differences in expression of miRNA between EAC and SCC.

In 2015, Chiam et al. [31] verified the presence of serum exosome miRNAs in 18 cases of locally advanced EAC, ten cases of BE, and 19 healthy controls, and identified a multi-biomarker panel (RNU6-1/miR-16-5p, miR-25-3p/miR-320a, let-7e-5p/miR-15b-5p, miR-30a-5p/miR-324-5p, and miR-17-5p/miR-194-5p) that is more specific than any single miRNA. They reported that this panel distinguished EAC patients from healthy controls and BE patients with high specificity and sensitivity (ROC = 0.99; 95% confidence interval (CI), 0.96–1.0).

Warnecke-Eberz et al. [32] reported overexpression of serum exosome-derived miR-223-5p and miR-483-5p in EAC patients. Furthermore, regarding differences in TNM classification, miR-223-5p expression was higher in T2 (*p* = 0.006) than in T3. These data suggest that miR-223 is secreted at high levels during the early stages of tumor development.

Bus et al. [23] compared BE, EAC, and control plasma miRNAs, and reported that miR-382-5p was significantly up-regulated, and miR-133a-3p was down-regulated, in EAC patients, and that combinations of three or more miRNAs could distinguish EAC from controls (AUC, 0.846) and BE from EAC (AUC, 0.797).

In 2016, Yan et al. [33] reported that exosomes extracted from EAC-conditioned medium or serum from EAC patients contained significantly higher levels of miR-93-5p and miR-21-5p than normal medium or normal serum. In addition, they cultivated normal enteroids and BE organoids for 3 weeks in EAC-conditioned medium. They found that EAC-conditioned medium induced dysplasia and proliferation of enteroids and BE organoids. Therefore, they hypothesized that exosomes from cancer cells play a role in induction of morphological changes in organoids and promote their growth.

Zhang et al. [34] compared circulating miRNAs in serum from EAC patients and healthy controls, and found that miR-25-3p and miR-151a-3p were significantly up-regulated, and that miR-100-5p and miR-375-3p were significantly down-regulated in EAC patients. 

Chen et al. [35] isolated exosomes from serum samples from EAC patients and controls, and reported that miR-21-5p, -16-5p, -25-3p, and -155-5p were significantly up-regulated, and miR-192-5p was significantly down-regulated, in EAC patients.

We [36] also analyzed tissue and blood miRNAs from human interleukin-1β (IL-1β) transgenic mice (L2-IL-1β mice) as a model of BE. We then compared expression of human plasma miRNAs in eight EAC patients, eight BE patients, and six healthy individuals using a public dataset. Compared with the BE group, levels of miR-128-3p and miR-328-3p were higher in the EAC group, and those of miR-143-3p, -144-3p, -15a-5p, -1-3p, and -133b were lower.

In 2018, Miyoshi et al. [37] profiled miRNAs expressed in 42 pairs of EAC and normal tissues, and validated them in 44 EAC, 20 BE, and 30 control sera. Six miRNAs (miR-106b-5p, -146a-5p, -15a-5p, -18a-5p, -21-5p, and -93-5p) were significantly up-regulated and could be used to distinguish EAC patients from healthy subjects (AUC, 0.86).

In 2019, Wang et al. [27] profiled BE and healthy subjects, and compared serum miRNAs among EAC, BE, and healthy groups. They found that miR130a-3p was significantly up-regulated in the EAC group, and that expression of miR130a-3p increased gradually from early (I, II) to advanced (III, IV) stage. As we mentioned in the BE section, they also showed that levels of miR130a-3p were significantly higher in patients with LGD and HGD BE than in those with metaplasia (sensitivity, 70.5%; specificity, 62.5%), suggesting that increased miR130a-3p expression in plasma may correlate with early EAC.

Craig et al. [38] compared miRNAs in serum from five controls, nine GERD, seven BE, five BE with LGD, and five EAC patients with those in biopsy tissues. Some characteristic miRNAs were identified in biopsy tissues, but there was no difference between expression of serum miRNAs between EAC patients and controls.

Fassan et al. [39] also compared serum miRNA from patients with BE, HG-IEN (dysplastic or intraepithelial neoplasia), and EAC. Seven (miR-92a-3p, -151a-5p, -362-3p, -345-3p, -619- 3p, -1260b, and -1276) were up-regulated in the HG-IEN/EAC samples compared with the non-dysplastic BE samples, whereas three (miR-381-3p, -502-3p, and -3615) were down-regulated.
jcm-10-00117-t002_Table 2Table 2MiRNAs used to detect EAC.Author, Year [Reference]Sample SizeSampleUp-RegulatedDown-RegulatedXie et al.2013 [30]SCC (*n* = 32)EAC (*n* = 4)Healthy (*n* = 19)Wholesaliva-10b-3p, -144-3p, -451a
Saliva supernatant-10b-3p, -144-3p, -21-5p, -451aChiam et al.2015 [31]EAC (*n* = 18)BE (*n* = 10)Healthy (*n* = 19)SerumRNU6-1/-16-5p,-25-3p/-320alet-7e-5p/-16b-5p,-30a-5p/-324-5p,-17-5p/-194-5p
Warnecke-Eberz et al.2015 [32]EAC (*n* = 59)Healthy (*n* = 4Serum-223-5p, -483-5p
Bus et al.2016 [23]EAC (*n* = 59)BE (*n* = 41)Healthy (*n* = 15)Plasma-382-5p-133a-3pYan et al.2016 [33]
Mediumserum-93-5p, -21-5p
Zhang et al.2016 [34]EAC (*n* = 10)Healthy (*n* = 11)Serum-25-3p, 151a-3p-100-5p, -375-3pChen et al.2017 [35]EAC (*n* = 9)Healthy (*n* = 9)Serum-21-5p, -16-5p, -25-3p, -155-5p-192-5pFassan et al.2017 [25]EAC (*n* = 30)Dyspeptic patients (*n* = 20)Atrophic gastritis (*n* = 15)BE (*n* = 15)Plasma-223-3p
Matsuzaki et al.2017 [36]EAC (*n* = 8)BE (*n* = 8)Healthy (*n* = 6)Plasma-128-3p, -328-3p-143-3p, -144-3p, -15a-5p, -1-3p, -133bMiyoshi et al.2018 [37]EAC (*n* = 44)BE (*n* = 20)Healthy (*n* = 30)Serum,-106b-5p, -93-5p, -146a-5p, -15a-5p, -18a-5p, -21-5p,
Wang et al.2019 [27]EAC (*n* = 40)BE (*n* = 60)Healthy (*n* = 30)Serum-130a-3p
Fassan et al.2020 [39]EAC (*n* = 8)BE (*n* = 12)HG-IEN (*n* = 5)Serum-92a-3p, -151a-5p, -362-3p, -345-3p, -619-3p, 1260b,-1276-381-3p, -502-3p, -3615EAC, esophageal adenocarcinoma; BE, Barrett’s esophagus; HGD, high-grade dysplasia.

### 3.3. MiRNAs Related to Prognosis and Treatment Responses in EAC Patients

There have been some attempts to use miRNAs not only for diagnosis of EAC but also for treatment responses and prognosis. With respect to recurrence, Gu et al. [40] investigated the role of serum miRNAs as predictors of recurrence. They profiled serum miRNAs from 72 EAC patients (32 with recurrence and 40 without) and validated them in 329 EAC patients (132 with recurrence and 197 without). The results showed that serum miR-331-3p was significantly down-regulated in patients with recurrence, making it a useful biomarker for identifying EAC patients at high risk of recurrence. The following four studies examined treatment response and prognosis.

Odenthal et al. [41] examined serum miRNAs in 50 patients with locally advanced EAC who underwent neoadjuvant chemotherapy and surgery, and assessed whether serum miRNAs were useful indicators of treatment response. No serum miRNAs were associated with a therapeutic response, but there was a significant correlation between expression of miR-192-5p and miR-222-3p and T classification, and between expression of miR-302c-3p and miR-222-3p and overall survival (OS). They reported that low miR-222-3p expression and high 302c-3p expression may suggest significantly better OS.

In 2014, Pu et al. [42] attempted to identify serum miRNAs as diagnostic, prognostic, and predictive biomarkers for EAC. When comparing miRNAs in the EAC and the control groups, they identified four (miR-126-3p, -142-3p, -331-3p, and -18a-5p) that were specific for EAC (*p* < 0.05). In addition, five miRNAs were significantly associated with survival, two of which were associated with recurrence (*p* < 0.05) (the specific miRNAs were not identified). Also, in cases with high miR-30c-5p expression, the risk of death (HR = 0.28, 95% CI = 0.12–0.65, *p* = 0.003) and recurrence (HR = 0.49; 95% CI = 0.24–0.99, *p* = 0.047) was lower than in cases with low expression (“risk of death” was not defined). Among 118 patients who received chemo/chemoradiation therapy (93% and 84% received 5FU and platinum, respectively), those in both treatment groups with high expression of miR-30c-5p had a reduced risk of mortality. In the platinum group, miR-26a-5p was significantly related to increased risk of death (HR = 5.78; 95% CI = 1.59–21.06, *p* = 0.008) and recurrence (HR = 7.76; 95% CI = 2.30–26.18, *p* = 0.001), and miR-142-3p was associated with an increased risk of recurrence (HR = 2.67; 95% CI = 1.09–6.56, *p* = 0.032). By contrast, in the 5FU group, miR-127-3p and miR-486-5p were significantly associated with both survival and recurrence (it is unclear whether high or low expression was important).

In 2015, Zhai et al. [43] investigated whether serum miRNA profiles could predict the prognosis of EAC together with *H. pylori* infection status. Serum samples from EAC patients and healthy individuals were used to profile miRNAs, and serum from other EAC patients were evaluated to examine the association between survival and the presence or absence of *H. pylori* infection (*H. pylori* was detected by immunoblotting). Overall, serum miRNAs were not significantly associated with the prognosis of EAC in the *H. pylori*-positive group. However, 16 miRNAs (miR-1253, -1273d, -187-5p, -1912, -200b-5p, -2276, -3147, -324-3p, -326, -3652, -3935, -4267, -4274, -4286, -4323, and -640) were significantly associated with OS (all *p* < 0.05) in the *H. pylori*-negative group. This suggests that miRNAs affect the survival rate of those with EAC and may be related to the presence or absence of *H. pylori* infection.

Petrick et al. [44] used US multicenter population-based serum samples from patients enrolled between 1993 and 1995 to compare miRNAs in EAC and healthy individuals, and to investigate survival rates. In the EAC group, 68 serum miRNAs were significantly up-regulated and 11 serum miRNAs were down-regulated. MiR-4253 and miR-1238-5p were associated with survival after EAC diagnosis (HR = 4.85, 95% CI, 2.30–10.23, BH-adjusted *p* = 0.04; and HR = 3.81, 95% CI, 2.02–7.19, BH-adjusted *p* = 0.04, respectively). This study suggests that circulating miRNAs may be associated with survival after EAC diagnosis.

## 4. Discussion

### 4.1. BE diagnosis

Seven miRNAs (miR-15a-5p, -130a-3p, -143-3p, -194-5p, 195a-5p, -215-5p, and -451a) were up-regulated and three miRNAs (miR-136-5p, -320e, and -199a-3p) were down-regulated. MiR-15a-5p [37], -130-3p [27], and -451a [30] are also up-regulated in EAC, whereas miR-143 is down-regulated.

MiR-15a-5p is involved in colon cancer, cervical cancer, and cervical cell cancer [45,46,47,48], and the main function of miR-130a-3p is to inhibit tumor progression [49,50]. MiR-143-3p [51,52,53] and miR-451a are tumor suppressors that correlate with non-small cell lung cancer (NSCLC) and colorectal cancer [53,54]. MiR-194-5p inhibits cell migration and invasion during gastric cancer progression by down-regulating FoxM1 [55]. MiR-136-5p suppresses lung adenocarcinoma [56] and renal cell cancer [57]. MiR-199a-3p suppresses EAC [58] and other cancers such as gastric cancer [59]. A report suggests that tissue miR-215-5p is specific for BE [60]; however, others suggest that it suppresses some tumors [61,62]. Previous studies show that expression of miR-320e is higher in EAC tissues than in BE tissues; high miR-320e tissue levels are associated with an adverse prognosis in those with colon cancer [63], although the underlying mechanism is unclear (Table 3).

Based on the above, levels of miRNA in blood differ between BE and healthy subjects; two papers report upregulation of miR-194-5p in BE [23,24]. MiR-194-5p and miR-215-5p levels are higher in those with BE, and even in those with CLE, than in controls [24]. It is thought that specific serum miRNAs start to rise at the very early stage, when the SE begins to show morphological changes, and that this may be useful for diagnosis of early dysplasia and BE.

### 4.2. EAC Diagnosis

Many miRNAs are associated with EAC diagnosis, among which miR-21-5p, -25-3p, and -93-5p are mentioned in more than one paper. MiR-21-5p was reported in four papers [30,33,35,37]; indeed, it is one of the most commonly observed aberrant miRNAs in human cancer. Numerous studies show that miR-21-5p promotes carcinogenesis and tumor progression by targeting genes encoding PDCD4, PTEN, BTG2, FasL, IGFBP3, TGF-β1, FBXO11, and TIMP3 [66,67,68,69,70,71]. Many studies explored the function of miR-21-5p with respect to prognosis and diagnosis. With respect to EAC, miR-21-5p upregulation promotes metastasis of EAC by regulating cancer cell apoptosis [72]; it is also an independent prognostic biomarker for disease-specific survival [73]. More than two papers report upregulation of miR-25-3p [31,34] and miR-93-5p [33,37]. High expression of miR-25-3p in EAC tissues is reported in several papers; this miRNA promotes cell migration and invasion of SCC by targeting CDH1 [74]. Mir-93-5p was identified by studies examining EAC tissue miRNA; the combination of miR-25/miR-93 establishes an immunosuppressive microenvironment in breast cancer by inactivating the cGAS pathway [75], and it also reported to have the promotion function on cervical cancer by targeting THB2/Matrix metalloproteinases (MMPS) pathway [76] MiR-15a-5p is both up-regulated [37] and down-regulated [36,77] in cancer serum; some reports suggest that it acts as a tumor suppressor [47,78], whereas some suggest that it acts as a tumor promoter [48]. Further research is necessary. MiR-223-5p is expressed at higher levels at T2 than at T3. MiR-223-5p inhibits tumor progression [79], whereas miR-130a-3p expression increases as the tumor stage progresses [27], suggesting that it promotes tumor growth [64] (Table 4).

Many studies show that extracellular miRNA levels are significantly different between EAC, BE, and controls, but there is also a report that the utility of serum miRNAs is limited [38]. The roles of miR-21-5p, -25-3p, and -93-5p in cancer have been reported in multiple papers [76]. In addition, serum miR-223-5p levels increase more quickly at the early stage of EAC [32], and serum levels of miR-130a increase as tumor stage progresses [27]. Thus, miRNAs are useful markers for early detection and staging of EAC.

### 4.3. EAC Recurrence

Two papers reported that serum levels of miR-331-3p, -30c-5p, 142-3p, 127-3p, 486-5p, and -26a-5p correlate with recurrence of EAC [40,42]. There are few reports of cellular or extracellular miRNAs in the context of EAC recurrence. Matsui et al. [80] identified miR-652-5p, -7-2-3p, -3925-3p, and -219-3p as being prognostic for recurrence in patients with stage II/III EAC. But in the two papers we found here, the same extracellular miRNAs were not reported.

MiR-142-3p suppresses growth of colorectal cancer [81] and breast cancer [82] cells by targeting CDK4, Beach-1, and CDC25C [83]. MiR-331-3p is useful for diagnosis of EAC [42] and is known to promote tumor progression [83]. As we mentioned in the previous paragraph about EAC recurrence, four (miR-30c-5p, -127-3p, -486-5p, and -26a-5p) serum miRNAs are associated with OS/prognosis (Table 5).

### 4.4. Survival and Prognosis

Among the miRNAs associated with OS or prognosis, miR-30c-5p, -26a-5p, -127-3p, and -486-5p are also associated with recurrence. Serum miR-222-3p is also associated with T classification [41,42].

MiR-30c-5p suppresses tumorigenesis in gastric cancer [89] and renal cell carcinoma (RCC) [90]. MiR-331-3p regulates cancer growth by targeting HER2, overexpression of which promotes cancer cell proliferation, resistance to apoptosis, angiogenesis, and metastasis [95]. MiR-331-3p expression is down-regulated in various cancers (e.g., cervical cancer [84], colorectal cancer [85], ovarian cancer [86], gastric cancer [87], prostate cancer, and glioblastoma). MiR-26a-5p inhibits migration and invasion of cancer cells by targeting the TMEM184B gene [88]. MiR-127-3p acts as a tumor growth suppressor by targeting KIF3, ITGA6, and BAG5 in SCC [96], osteosarcoma [92], and ovarian cancer [93]. MiR-486-5p stimulates cell proliferation, migration, and invasion by inhibiting PTEN expression and by activating the oncogenic PI3K/AKT pathway [97,98], although one study concluded that it works to suppress tumors [99]; further verification is required (Table 5).

In addition, Zhai et al. reported that miRNAs that predict prognosis differed according to *H. pylori* infection status [43]. *H. pylori* infection induces chronic gastritis and metaplasia, which can alter miRNA expression profiles in human gastric mucosa [100,101]. Wang et al. reported that miRNA expression in 53 gastric cancer tissues differed between *H. pylori*-positive and *H. pylori*-negative patients [102]. Importantly, most patients with EAC do not have *H. pylori* infection because *H. pylori* infection reduces gastric acid secretion and acid regurgitation into the esophagus. [103]. Differences in expression of miRNAs associated with the prognosis of EAC might be due to different etiologies of EAC.

From the above, we can see that many miRNAs are associated with tumor progression. Thus, increased or decreased levels of these in the blood may reflect the growth rate of the tumor, and will therefore reflect survival or recurrence rates.

### 4.5. Potential of Extracellular miRNAs as BE/EAC Diagnosis/Prediction and Prognosis Markers

Detection of EAC by endoscopy requires skill and continuous training. In Western countries, although several screening methods for EAC have been suggested, the most cost-effective strategies for risk stratification and surveillance intervals remain controversial [17]. In fact, the detection rate of EAC by surveillance alone is low; less than 15% of all EAC cases are detected by endoscopic surveillance [104]. A ‘liquid biopsy’, such as circulating tumor DNA (ctDNA) and miRNA, has the potential to detect EAC at the early stage without interobserver variability [105]. Sampling biofluid is quicker and less invasive than obtaining cancer tissues, and it is not affected by the anatomical location of the cancer. However, the diagnostic accuracy, cost, and reproducibility must be improved [106,107]. In particular, altered expression of miRNA detected by one method is sometimes not validated by another method. Therefore, careful validation using multiple orthogonal methods is important to check the robustness of findings.

It seems unlikely that a satisfactory detection rate will be achieved by testing for just one miRNA, as various disease conditions can alter extracellular miRNA profiles in the body. However, by combining multiple miRNAs, disease-specific screening may be possible. At present, several companies worldwide are working to develop extracellular miRNA-based diagnostics. Two case-control studies (NCT02464930 and NCT02812680) are registered, both of which are recruiting about 200 participants. However, no population-based study has been performed. We also reported that several cancer types, such as breast and ovarian cancer, can be diagnosed at the early stage by analyzing serum miRNAs [108,109]. However, we also need to identify the underlying biological mechanisms by which these miRNA profiles change under different disease conditions, a process that will take several years. In terms of the cost, it should not be a prohibitive factor for clinical implementation because, if this technology becomes popular, the cost will fall.

## 5. Conclusions

Many miRNAs have been identified in the context of BE/EAC diagnosis and prognosis. In particular, several studies identified the same serum miRNAs (miR-21-5p, miR-25-3p, and miR93-5p) as being associated with EAC. However, since these studies are small-scale, and the sample types, method of miRNA analysis, pathological definition of disease, study populations, and outcomes vary, it is hard to identify useful miRNAs. To bring miRNA analysis into clinical practice, we must overcome many challenges, including standardization of methodologies, improvement of diagnostic performance, and cost-effectiveness [107]. Further investigations are warranted to firmly establish extracellular miRNAs as useful clinical biomarkers.

## Figures and Tables

**Figure 1 jcm-10-00117-f001:**
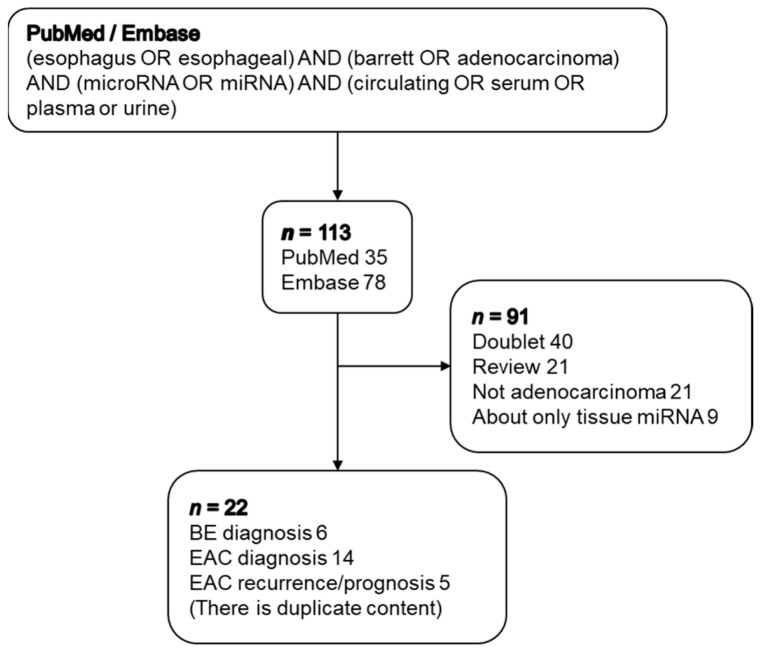
Flow chart showing the selection process.

**Figure 2 jcm-10-00117-f002:**
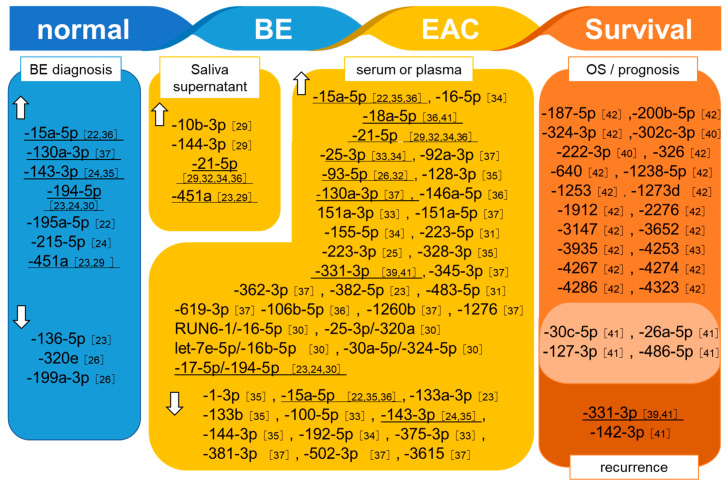
List of dysregulated miRNAs (identified in serum or saliva) that are specific for BE or EAC. Superscript numbers indicate references. Underlined miRNAs were reported in multiple reports. MiRNAs associated with both OS/prognosis and recurrence are shown as a Venn diagram.

**Table 3 jcm-10-00117-t003:** Functional roles and validated targets of miRNAs associated with BE diagnosis.

miRNA	Functional Role	Target	Cancer Type	Sample
miR-15-a-5p	Tumor promoter Tumor suppressor	TP53INP1 [48] WNT3A [47]	Cervical cancer Endometrial Cancer	30 tissue 8 tissue
miR-130a-3p	Inhibit cell proliferation, migration, and invasion	RAB5B [50] PTEN, p21 [64]	Breast cancer Gastric cancer	40 tissue 30 tissue
miR-143-3p	Tumor suppressor	TAK1 [51] MAPK7 [52] KRAS [53]	Ovarian cancer Breast cancer PDAC	4 cell lines 37 tissue
miR-215-5p	Tumor suppressor	EGFR [61] Sox9 [62]	Colorectal cancer Breast cancer	252 tissue 39 tissue
miR-194-5p	Inhibiting cell migration and invasion during cancer progression	FoxM1 [55]	Gastric cancer	50 tissue
miR-451a	Inhibits the migratory and invasive abilities	ATF2 [65]	NSCLC	55 tissue
miR-199a-3p	Prognostic indicator Tumor suppressor	ETNK1 [59] PAK4 [58]	Gastric cancer Esophageal cancer	39 tissue

BE, Barrett’s esophagus; PDAC, pancreatic ductal adenocarcinoma; NSCLC, non-small cell lung cancer.

**Table 4 jcm-10-00117-t004:** Functional roles and validated targets of miRNAs associated with EAC diagnosis.

miRNA	Functional Role	Target	Cancer Type	Sample
miR-21-5p	Promotes anoikis resistance and metastasis	TGFβ1 [68] TIMP3 [69] IGFBP3 [67] FBXO11 [71]	NSCLC RCC GBM	104 tissue
miR-25-3p	Promote migration invasion	CDH1 [74]	SCC	
miR-93-5p	Promote progression	PTEN [75] THBS2/MMPS [76]	Breast cancer Cervical cancer	28 tissue
miR-15a-5p	Tumor suppressor Promotor	E2F1, MYCN [78] TP53INP1 [48]	Neuroblastoma Cervical cancer	30 tissue

NSCLC: non-small-cell lung carcinoma, RCC: renal cell carcinoma, GBM: glioblastoma multiform, SCC: squamous cell carcinoma.

**Table 5 jcm-10-00117-t005:** Functional roles and validated targets of miRNAs associated with overall survival, prognosis, and recurrence.

miRNA	Functional Role	Target	Cancer Type	Sample
miR-331-3p	Tumor suppressor	HER2 [84,85] RCC2 [86] E2F1 [87]	Pancreatic cancer Colorectal cancer Cervical cancer Ovarian cancer Gastric cancer	44 plasma 29 tissue 20 tissue
miR-26a-5p	Tumor suppressor	TMEM184B [88]	OSCC	36 tissue
miR-30c-5p	Tumor suppressor	MTA1 [89] HSPA5 [90]	Gastric cancer RCC, Bladder cancer, Prostate cancer	40 tissue 130 urine
miR-127-3p	Tumor suppressor	KIF3B [91] ITGA6 [92] BAG5 [93]	OSCC Osteosarcoma Ovarian cancer	45 tissue 20 tissue
miR-142-3p	Tumor suppressor	CDK4 [81] CDC25C [94] Beach-1 [82]	Colorectal Cancer Breast Cancer	116 tissue 42 tissue

OSCC: oral squamous cell carcinoma, RCC: renal cell carcinoma.

## Data Availability

No new data were created or analyzed in this study. Data sharing is not applicable to this article.

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
