# Peer review of "Extracellular miRNAs for the Management of Barrett’s Esophagus and Esophageal Adenocarcinoma: A Systematic Review"

_jcm, 2020, doi:10.3390/jcm10010117_

Round 1

Reviewer 1 Report

1. Authors have done the hard work to summarize everything related to the topic in one article and that is what makes it useful.

Unfortunately there is a lot of heterogeneity among the articles in terms of which MiRNA is tested, which population (BE, EAC) and to who it is compared, what is the outcome tested (diagnosis, treatment response, prognosis)- which limits the ability to actually stack up results together and generate conclusions bigger/ more meaningful than the individual ones.

But, overall, it is a field in evolution and putting everything together gives a perspective for future research goals.

2. In conclusion, authors should write/recommend future research goals-

-At this stage, the sensitivity and specificity are not high enough to replace current standards but may be combining few markers may help us achieve that.

- Also the financial aspect of this kind of testing needs to be evaluated in clinical side, before even recommending it

3. Minor language editing-

-line 87 to 94- needs to be deleted. 

-Also in line 87- venn diagram (needs to be written in complete)

Author Response

1. Authors have done the hard work to summarize everything related to the topic in one article and that is what makes it useful.
Unfortunately there is a lot of heterogeneity among the articles in terms of which MiRNA is tested, which population (BE, EAC) and to who it is compared, what is the outcome tested (diagnosis, treatment response, prognosis)- which limits the ability to actually stack up results together and generate conclusions bigger/ more meaningful than the individual ones.
But, overall, it is a field in evolution and putting everything together gives a perspective for future research goals.

REPLY:
I appreciate your comments and advice. I completely agree with you. Collected literature here used different sample types, such as serum, plasma, and saliva, different pathological definitions, different populations, and different experimental methods for measuring miRNAs. So, we cannot conclude some specific miRNAs are useful, but we can say that pursuing investigations of extracellular miRNAs as biomarkers are meaningful. We added the sentences about this in the Conclusion section:

“However, since sample types, methods for miRNA analysis, the pathological definition of diseases, study population, and outcomes vary among reports, it is still hard to specify the name of useful miRNAs. To bring miRNAs into clinical practice, there are many challenges to overcome, such as standardization of methodologies, improvement of the diagnostic performance, and cost-effectiveness [106]. Further investigations are warranted to establish extracellular miRNAs as clinical biomarkers.”

2. In conclusion, authors should write/recommend future research goals-
-At this stage, the sensitivity and specificity are not high enough to replace current standards but may be combining few markers may help us achieve that.
- Also the financial aspect of this kind of testing needs to be evaluated in clinical side, before even recommending it

REPLY:
Thank you for your constructive suggestion. I newly added discussion section 4.5. and discuss what you pointed out as follows:

3. Minor language editing-
-line 87 to 94- needs to be deleted.
-Also in line 87- venn diagram (needs to be written in complete)

REPLY:
We greatly appreciate your pointing out. We revised them.

Reviewer 2 Report

Major comments

  • The authors set out to summarize the knowledge regarding the potential utility of miRNAs for improved detection and prognosis monitoring in EAC, and while they did successfully list a lot of significant miRNA alterations, I feel that more discussion could have been given to whether these significant/validated miRNAs have any real clinical utility (or will in future). Serum markers with prognostic ability are the holy grail in EAC, but is there evidence to suggest that miRNAs could be used reliably in the clinical setting? Are the data robust enough – have any clinical trials been carried out? Are any companies working on this? Would be good to include this information, as that is what I thought I would learn from this review.
  • Was the same/similar methodology used to measure/quantify miRNAs between papers, and was this found to be comparable between studies? It would be informative if the authors could give more detail on how miRNAs are detected. Is the methodology costly? And could this potentially be a prohibitive factor for clinical implementation?
  • It would be good if the authors could include information on cohort sizes for all studies, to give the reader an idea of the robustness of each study cited. Would be good to indicate for example in Tables 4 and 5, the total number of patient samples tested, perhaps in brackets after the cancer type, to give the reader a sense of whether these markers have been validated in 100s or 1000s of patients. How new is this technology?
  • The authors mention differences noted in miRNA levels between those with and without H pylori infection. It would be interesting to comment further on this, and perhaps other factors which may confound miRNA analysis or make it otherwise less accurate. Is the mechanism of action of H pylori interaction with miRNAs known?
  • The authors mention different sample types were used (e.g. plasma, serum, saliva) - are there any benefits to using one type over another that again may encourage/discourage clinical use of miRNA analysis? Would be good to include discussion of this.

Minor comments/typos

Abstract line 16 – Is “systemically” or “systematically” meant?

“In 2012, around 52,000 people developed EAC worldwide” – could choose a more recent paper

“according to the American Society of Clinical Oncology, 18,440 people have been newly diagnosed and 16,000 have died from esophageal cancer in the United States in 2020.” – 2020 is probably not the best representative year for new diagnoses as many cancer centres are reporting significantly fewer new diagnoses than usual, with many countries in quarantine. No citation included here – a registry report would be preferable.

“mainly due to the decreased prevalence of H. pylori infection” – insert citation

Line 39 – remove space after “[7-9] .”

Moving Figure 1 into section 2.1 and away from Figure 2 would improve the flow.

Line 86, letter missing? Template text erroneously included here, please remove and replace with further explanation of Figure 2, specifically the significance of the color coding approach used.

Line 103 – Delete space after “p= ”

Line 163 – “Th” should be “The”?

Line 248-249 – Citation needed and need to state control for up/downregulation.

Line 278 – Delete unnecessary gaps

Line 303 – Do you mean “in the previous sentence” or in the previous section? The miRNAs mentioned don’t appear to correlate with either, so perhaps clarify this sentence. The subsequent section, perhaps is meant?

Line 325 – “In particular, several studies have identified the same serum miRNAs”

Author Response

Major comments
• The authors set out to summarize the knowledge regarding the potential utility of miRNAs for improved detection and prognosis monitoring in EAC, and while they did successfully list a lot of significant miRNA alterations, I feel that more discussion could have been given to whether these significant/validated miRNAs have any real clinical utility (or will in future). Serum markers with prognostic ability are the holy grail in EAC, but is there evidence to suggest that miRNAs could be used reliably in the clinical setting? Are the data robust enough – have any clinical trials been carried out? Are any companies working on this? Would be good to include this information, as that is what I thought I would learn from this review.

REPLY:
Thank you for your important suggestions. To be honest, we think that it will take a further long time to obtain sufficient robust clinical evidence and to reveal underlying biological mechanisms why these miRNA profiles are changed by disease conditions. We know several companies are working on this but it would not be fair to mention some names of companies here. In the manuscript, we added a new section 4.5. to describe that miRNA diagnostics are truly promising but there are still various issues to be overcome.

Was the same/similar methodology used to measure/quantify miRNAs between papers, and was this found to be comparable between studies? It would be informative if the authors could give more detail on how miRNAs are detected. Is the methodology costly? And could this potentially be a prohibitive factor for clinical implementation?

REPLY:
This is also a very good question. In fact, we experienced that some miRNA can be detected by one method but cannot by the other method. This is why the validation using multiple orthogonal quantification methods is important to check the robustness of findings. In terms of the cost, we do not think it a prohibitive factor for clinical implementation. If this technology becomes popular, the cost must be gradually decreased. These discussions are also included in a new section 4.5.

It would be good if the authors could include information on cohort sizes for all studies, to give the reader an idea of the robustness of each study cited. Would be good to indicate for example in Tables 4 and 5, the total number of patient samples tested, perhaps in brackets after the cancer type, to give the reader a sense of whether these markers have been validated in 100s or 1000s of patients. How new is this technology?

REPLY:
We added a new column in Tables 4 and 5 to describe sample size to reveal functional roles/target mRNAs in previous papers. Blanks mean that these are analyzed only by cell lines.

The authors mention differences noted in miRNA levels between those with and without H pylori infection. It would be interesting to comment further on this, and perhaps other factors which may confound miRNA analysis or make it otherwise less accurate. Is the mechanism of action of H pylori interaction with miRNAs known?

REPLY:
We added the following paragraph in discussion section 4.4.

“In addition, Zhai et al reported that miRNAs to predict prognosis were different depending on the H. pylori infection status [43]. H. pylori infection induces chronic gastritis and metaplasia, which can change miRNA expression profile in human gastric mucosa [100][101]. In gastric cancer tissues, Wang et al. reported that fifty-three miRNA expression was different between H. pylori-positive and H. pylori-negative [102]. Importantly, most patients with EAC do not have H. pylori infection because H. pylori infection can reduce gastric acid secretion and acid regurgitation to the esophagus [103]. Differences of miRNAs associated with the prognosis of EAC might be due to the different etiology for the development of EAC.”

The authors mention different sample types were used (e.g. plasma, serum, saliva) - are there any benefits to using one type over another that again may encourage/discourage clinical use of miRNA analysis? Would be good to include discussion of this.

REPLY:
I really appreciate your advice. I added the text about sample types in discussion section 4.5.

Minor comments
Abstract line 16 – Is “systemically” or “systematically” meant?
REPLY:
Thank you for your notification. I changed here ‘systematically’

“In 2012, around 52,000 people developed EAC worldwide” – could choose a more recent paper

REPLY:
Thank you for your advice. We replaced the following sentence: “1 in 100 men in the Netherlands and United States are predicted to be diagnosed with EAC during their lifetime by 2030”.

“according to the American Society of Clinical Oncology, 18,440 people have been newly diagnosed and 16,000 have died from esophageal cancer in the United States in 2020.” – 2020 is probably not the best representative year for new diagnoses as many cancer centres are reporting significantly fewer new diagnoses than usual, with many countries in quarantine. No citation included here – a registry report would be preferable.

REPLY:
We agree that 2020 is not a representative year. Instead, we added regarding the prediction of EAC incidence as mentioned above.

“mainly due to the decreased prevalence of H. pylori infection” – insert citation

REPLY:
Thank you for pointing this out. We explained this more clearly with citations as follows:”The increase of EAC can be explained by the decreased prevalence of Helicobacter pylori (H. pylori) infection, although the relationship between H. pylori and EAC occurrence is still controversial. [8-10].”

Line 39 – remove space after “[7-9] .”

REPLY:
We deleted this blank.

Moving Figure 1 into section 2.1 and away from Figure 2 would improve the flow.

REPLY:
We revised as you commented.

Line 86, letter missing? Template text erroneously included here, please remove and replace with further explanation of Figure 2, specifically the significance of the color coding approach used.

REPLY:
We sincerely apologize this error. We deleted the template and added the following figure legend: “MiRNAs shown in red were reported as dysregulated in at least two papers, whereas those in white are reported to be biomarkers for multiple purposes. MiRNAs associated with both OS/prognosis and recurrence are listed as a Venn diagram.”

Line 103 – Delete space after “p= ”

REPLY:
Thank you so much for detecting it. We deleted the space.

Line 163 – “Th” should be “The”?

REPLY:
Thank you very much. We corrected the spell.

Line 248-249 – Citation needed and need to state control for up/downregulation.

REPLY:
This part is just a summary of the results. The details are described in the section 3.1.

Line 278 – Delete unnecessary gaps

REPLY:
Thank you for your pointing this out. I deleted the spaces.

Line 303 – Do you mean “in the previous sentence” or in the previous section? The miRNAs mentioned don’t appear to correlate with either, so perhaps clarify this sentence. The subsequent section, perhaps is meant?

REPLY:
We apologize for this wrong description. We changed it to "in the previous paragraph about EAC recurrence".

Line 325 – “In particular, several studies have identified the same serum miRNAs”

REPLY:
Thank you so much for correcting the grammar. I added ‘the’ in the sentence.

Round 2

Reviewer 2 Report

MDPI comments for authors

The authors have done a good job of addressing my concerns with this manuscript. However, I still feel that some further changes would greatly enhance the quality of this review and its usefulness in the field.

Specifically:

  • Tables 1-3 would also benefit from an indication of cohort numbers used in studies, as has been done for Tables 4&5. This information could be added without increasing the number of columns by combining Author and Year into a single ‘Reference’ heading.

  • Figure 2 – By putting citations in brackets after each miRNA to denote number of publications, you can eliminate the need for colour coding words, which can make the text difficult to read.

  • Section 4.5 contains a number of grammatical errors and missing words (e.g. “which would take than a few years.”) which affect the readability of the text.

  • Figure 2 is a useful summary figure – authors might consider moving it to a later part of the review, once the miRNAs contained within have already been mentioned in the text.

  • The sample numbers included in Tables 4+5 are surprisingly low for validation studies. Have any clinical trials been conducted in this area? (Please mention if so). Or is this area just very new?

  • The first letter of the Conclusion section text is appearing strangely on my screen – perhaps a formatting issue here?

Author Response

We truly thank your constructive suggestions. We highlighted the revised words and sentences in the manuscript. We hope this revised version would fulfill your requests.

1. Tables 1-3 would also benefit from an indication of cohort numbers used in studies, as has been done for Tables 4&5. This information could be added without increasing the number of columns by combining Author and Year into a single ‘Reference’ heading.

REPLY:
We added sample numbers and references in Table1-3. You must find these studies are small-scale studies. This issue was commented on in the conclusion.

2. Figure 2 – By putting citations in brackets after each miRNA to denote number of publications, you can eliminate the need for colour coding words, which can make the text difficult to read.

REPLY:

We added references and unified the letter color.

3. Section 4.5 contains a number of grammatical errors and missing words (e.g. “which would take than a few years.”) which affect the readability of the text.

REPLY:

The manuscript was reviewed by the professional English editing service.

4. Figure 2 is a useful summary figure – authors might consider moving it to a later part of the review, once the miRNAs contained within have already been mentioned in the text.

REPLY:

Thank you for your good suggestions. We moved figure2 just before the discussion section.

5. The sample numbers included in Tables 4+5 are surprisingly low for validation studies. Have any clinical trials been conducted in this area? (Please mention if so). Or is this area just very new?

REPLY:

By searching ClinicalTrial.gov, we identified two registered case-control studies (NCT02464930 and NCT02812680). Both studies are recruiting about 200 participants. No population-based study has been performed. This point was added in section 4.5.

6. It would be good if the authors could include information on cohort sizes for all studies, to give the reader an idea of the robustness of each study cited. Would be good to indicate for example in Tables 4 and 5, the total number of patient samples tested, perhaps in brackets after the cancer type, to give the reader a sense of whether these markers have been validated in 100s or 1000s of patients. How new is this technology?

REPLY:

We added sample sizes in Table 1-3. This clearly indicates that a large-scale study has not been conducted.

7.The first letter of the Conclusion section text is appearing strangely on my screen – perhaps a formatting issue here?

REPLY:

We could not find any problem here but we fixed the font type for this part.